# Trend analysis of mortality rates and causes of death in children under 5 years old in Beijing, China from 1992 to 2015 and forecast of mortality into the future: an entire population-based epidemiological study

Han Cao,[1,2] Jing Wang,[1,3] Yichen Li,[3] Dongyang Li,[3] Jin Guo,[1] Yifei Hu,[4] Kai Meng,[5] Dian He,[1] Bin Liu,[1,2] Zheng Liu,[1,2] Han Qi,[1,2] Ling Zhang[1,2]

HC and JW contributed equally.

For numbered affiliations see end of article.

**Correspondence to**
Proffesor Ling Zhang;
zlilyepi@ccmu.edu.cn

## ABSTRACT

**Objectives** To analyse trends in mortality and causes of death among children aged under 5 years in Beijing, China between 1992 and 2015 and to forecast under-5 mortality rates (U5MRs) for the period 2016–2020.

**Methods** An entire population-based epidemiological study was conducted. Data collection was based on the Child Death Reporting Card of the Beijing Under-5 Mortality Rate Surveillance Network. Trends in mortality and leading causes of death were analysed using the $\chi^2$ test and SPSS 19.0 software. An autoregressive integrated moving average (ARIMA) model was fitted to forecast U5MRs between 2016 and 2020 using the EViews 8.0 software.

**Results** Mortality in neonates, infants and children aged under 5 years decreased by 84.06%, 80.04% and 80.17% from 1992 to 2015, respectively. However, the U5MR increased by 7.20% from 2013 to 2015. Birth asphyxia, congenital heart disease, preterm/low birth weight and other congenital abnormalities comprised the top five causes of death. The greatest, most rapid reduction was that of pneumonia by 92.26%, with an annual average rate of reduction of 10.53%. The distribution of causes of death differed among children of different ages. Accidental asphyxia and sepsis were among the top five causes of death in children aged 28 days to 1 year and accident was among the top five causes in children aged 1–4 years. The U5MRs in Beijing are projected to be 2.88‰, 2.87‰, 2.90‰, 2.97‰ and 3.09‰ for the period 2016–2020, based on the predictive model.

**Conclusion** Beijing has made considerable progress in reducing U5MRs from 1992 to 2015. However, U5MRs could show a slight upward trend from 2016 to 2020. Future considerations for child healthcare include the management of birth asphyxia, congenital heart disease, preterm/low birth weight and other congenital abnormalities. Specific preventative measures should be implemented for children of various age groups.

## Strengths and limitations of this study

► Our study revealed the long-term trends and characteristics in mortality rates and leading causes of death in under-5 children through a 24-year analysis.
► We reported on the health conditions of children aged under 5 years following the implementation of the 'Selective Two-child Policy' in Beijing in 2014.
► The accuracy of the data is reliable because the sample comprises the entire population of all under-5 children in all districts of Beijing based on the surveillance network.
► Owing to the unique regional divisions in Beijing and lack of specific data on each district, we did not analyse the differences between rural and urban areas.
► The autoregressive integrated moving average (ARIMA) (1,1,1) model forecasted future U5MRs reliably for only a short period of time, because the model only considered variations in mortality with time instead of other possible impacting factors, and data have to be continually updated to predict further mortality rates. Other models which can predict U5MRs more accurately and capture its patterns more specifically need to be explored.
► Because we only collected 2-year data (2014 and 2015) after the 'Selective Two-child Policy', the representativeness of the changing trend and solid statistical analysis were limited. The relation between the changing trend in U5MRs and policy regulation is still unclear and needs to be observed for further years. We could only give a suggestion to raise healthcare workers' awareness about this potential problem.

## INTRODUCTION

The under-5 mortality rate (U5MR) is an important indicator that reflects the health of children and development of the economy

and culture of a country or region.[1] Millennium Development Goal 4 (MDG4) calls for the U5MR to be reduced by two-thirds between 1990 and 2015.[2] China established the National Maternal and Child Mortality Surveillance system (MCMS) in 1996 and has made considerable progress in reducing the U5MR over the past few decades.[3] With respect to the nationwide U5MR, MDG4 was achieved in 2008, 7 years in advance of the target.[4] Furthermore, the U5MR in China declined to 10.7‰ in 2015, a reduction of 80.1% from 1990.[5] Beijing established the Beijing Under-5 Mortality Rate Surveillance Network in 1992. However, no report on long-term dynamic trends in the U5MR has been produced since the establishment of this monitoring network.

The distribution of the causes of death is influenced by development of the social economy and healthcare.[6] Some changes have been noted in the nationwide distribution of the causes of death; pneumonia has dropped from the leading cause to fifth place whereas congenital heart disease climbed from sixth place to third place from 1996 to 2015.[3 6 7] Beijing, the capital city of China, is currently experiencing the most rapid developments in the economy and healthcare. However, the distribution of the causes of death in under-5 children has not been investigated within the last 24 years.

Furthermore, implementation of the 'Selective Two-child Policy' in 2014[8] and the 'Entire Two-child Policy' in 2016[9] in Beijing has led to a rise in the number of live births and older parturients.[10] Whether or not policy changes will contribute to an increase in the U5MR remains unknown. Therefore, it is worthwhile reporting the health conditions of under-5 children during this period and predicting future U5MRs.

The present study aimed to analyse dynamic changes in the U5MR and the distribution of the causes of death in Beijing, China from 1992 to 2015. The study also sought to establish a forecasting model to predict the U5MR for the following 5 years.

## METHODS

### Data sources

We used the surveillance data between 1992 and 2015 from the Beijing Under-5 Mortality Rate Surveillance Network and the Beijing Maternal and Child Health Care Information System, established in 1992 and 2003, respectively. The scope of monitoring covered all under-5 children in all 16 districts (Dongcheng, Xicheng, Haidian, Chaoyang, Fengtai, Shijingshan, Mentougou, Fangshan, Daxing, Shunyi, Huairou, Miyun, Yanqing, Changping, Pinggu and Tongzhou). Thus, this was an entire population-based epidemiological study. The surveillance network was rigidly implemented according to the Healthcare Operating Programme for Under-5 Children in Beijing and information on deaths was collected from the Child Death Reporting Cards through a three-level network approach (community/hospital level, district level and municipal level).

Stringent quality control was enforced at all three levels. Maternal and child healthcare institutions at the district level checked the Child Death Reporting Card within the jurisdiction on a quarterly basis and verifies the cards with the Centres for Disease Control and Prevention (CDC) at the district level. Furthermore, data from 20% of the communities and hospitals were extracted annually for quality control, inspection and investigation of missing reports. Moreover, maternal and child healthcare institutions at the municipal level checked the cards reported by the district on a quarterly basis, and extracted data collected annually from 20% of the districts to ensure data integrity and accuracy. In addition, data-related staff of local maternal and child healthcare facilities were trained by specialist groups at regular intervals. The annual live birth under-reporting rate was less than 1% and the under-reporting rate of deaths was less than 0.5% over the last 10 years in Beijing.

The main factor under surveillance was live births. A live birth was defined as a fetus born after 28 weeks of gestation (or with a birth weight >1000 g) with at least one of the following vital signs: heartbeat, breathing, pulsation of the umbilical cord or contraction of voluntary muscle. Only live births or the parents of those births on the census register in Beijing were included. From 1992 to 2015 a total of 1 881 952 live births and 12 541 deaths were recorded. Categorization of the causes of death was in accordance with the International Categorization of Diseases (ICD-10).

Because the data on the deaths of under-5 children in Beijing were derived from the Beijing Under-5 Mortality Surveillance Network, no further ethical approval was required for the present study.

### Procedures

We collected data from the Child Death Reporting Card of the Surveillance Network and Information System. The Child Death Reporting Card was in paperboard from 1992 to 2002, so we applied double entries and consistency checks in the Epidata 3.0 software to ensure the accuracy of the data. However, data collected between 2003 and 2015 were directly exported because electronic cards have been used since the information system was established in 2003, which ensured the accuracy of the data. We then extracted information on the child's name, gender, census register, date of birth, date of death and classification of any causes of death to establish a database in the Excel 2016 program. The number of live births recorded in the Beijing census register was exported directly from the Beijing Maternal and Child Healthcare Hospital.

### Statistical analysis

The trends in mortality rates and distribution of the causes of death were analysed with descriptive statistics and the $\chi^2$ test using SPSS 19.0 software. The neonatal mortality rate (NMR) was defined as the number of deaths within 28 days from birth per 1000 live births. The infant

mortality rate (IMR) was defined as the number of deaths within 1 year from birth per 1000 live births. The under-5 mortality rate (U5MR) was defined as the number of deaths within 5 years from birth per 1000 live births (as defined by the National Centre for Health Statistics). The annual average rate of reduction (AARR) was calculated as the average rate of change in mortality rates from 1992 to 2015 as follows:

$$AARR = \left( \sqrt[23]{\frac{MR_{2015}}{MR_{1992}}} - 1 \right) \times 100\%$$

U5MR was predicted by time series analysis. The U5MR was marked by various types of influencing factors, as well as continuity in time. The use of traditional regression analysis methods to predict its trends was challenging. Time series analysis uses the factor of time to replace other kinds of influencing factors. The autoregressive integrated moving average (ARIMA) model is one of the classic methods of time series analysis, based on past values of a series and previous errors in forecasting. Therefore, it can be appropriately used to forecast the U5MR. We considered the U5MRs from 1992 to 2013 as a training sample to fit the ARIMA model. The U5MRs from 1992 to 2013 and 2014–2015 were considered the internal and external samples for internal and external validation, respectively. Data analysis and establishment of the model was accomplished with EViews 8.0 and SPSS 19.0 software. Statistical significance was set at p<0.05.

## RESULTS

### Trends in mortality rates in under-5 children

The total number of deaths in neonates (<28 days), infants (<1 year) and under-5 children was 7619 (60.75%), 9836 (78.43%) and 12 541 (100%), respectively. Deaths in neonates accounted for more than 50% of all deaths among under-5 children during the 24 consecutive years under investigation (figure 1).

Table 1 summarises the mortality rates in neonates (NMR), infants (IMR) and under-5 children (U5MR) in Beijing, China from 1992 to 2015. All mortality rates exhibited an overall trend of significant decline ($\chi^2_{NMR} = 3200.17$, $P_{NMR} < 0.01$; $\chi^2_{IMR} = 3029.94$, $P_{IMR} < 0.01$; $\chi^2_{U5MR} = 4514.08$, $P_{U5MR} < 0.01$), with reductions by 84.06%, 80.04% and 80.17% during the period under investigation. The NMR had the highest AARR (7.67%) from 1992 to 2015 compared with the IMR (6.76%) and U5MR (6.79%). However, increases were observed in mortality rates from 2002 to 2003 (NMR, 5.22%; IMR, 17.04%; and U5MR 11.13%) and from 2014 to 2015 (NMR, 5.33%; IMR, 6.87%; and U5MR 6.92%).

### Distribution of the leading causes of death in under-5 children

There were 35 species of nine types causes of death for under-5 children under the surveillance in Beijing. Table 2 summarises mortality rates for the leading causes of death in under-5 children from 1992 to 2015. The

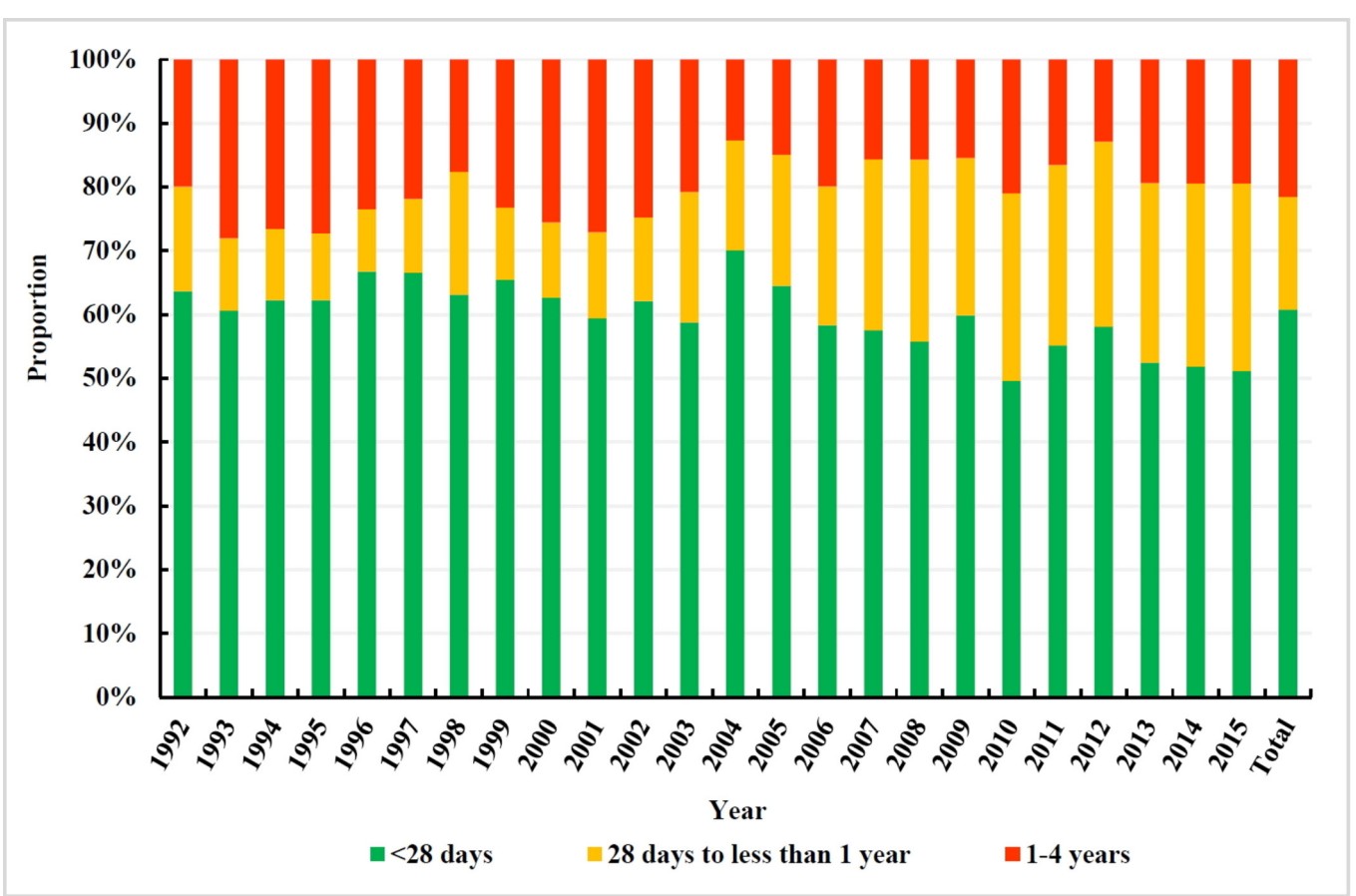

**Figure 1** Age composition of the death in children under 5 years old in Beijing, China, 1992–2015.

**Table 1** Mortality rates in neonates, infants and children under 5 years old in Beijing, China, 1992–2015

| Year | Live births | Neonates | | Infants | | Under-5 years | |
|---|---|---|---|---|---|---|---|
| | | Deaths | Mortality rate | Deaths | Mortality rate | Deaths | Mortality rate |
| 1992 | 82 105 | 814 | 9.91 | 1023 | 12.46 | 1278 | 15.57 |
| 1993 | 72 959 | 661 | 9.06 | 784 | 10.75 | 1090 | 14.94 |
| 1994 | 72 728 | 578 | 7.95 | 682 | 9.38 | 929 | 12.77 |
| 1995 | 71 049 | 547 | 7.70 | 639 | 8.99 | 879 | 12.37 |
| 1996 | 65 625 | 506 | 7.71 | 580 | 8.84 | 758 | 11.55 |
| 1997 | 66 230 | 487 | 7.35 | 572 | 8.64 | 732 | 11.05 |
| 1998 | 53 429 | 326 | 6.10 | 425 | 7.95 | 516 | 9.66 |
| 1999 | 53 123 | 330 | 6.21 | 387 | 7.28 | 504 | 9.49 |
| 2000 | 54 513 | 265 | 4.86 | 315 | 5.78 | 423 | 7.76 |
| 2001 | 48 443 | 224 | 4.62 | 275 | 5.68 | 377 | 7.78 |
| 2002 | 49 442 | 218 | 4.41 | 264 | 5.34 | 351 | 7.10 |
| 2003 | 38 400 | 178 | 4.64 | 240 | 6.25 | 303 | 7.89 |
| 2004 | 44 572 | 171 | 3.84 | 213 | 4.78 | 244 | 5.47 |
| 2005 | 58 409 | 207 | 3.54 | 273 | 4.67 | 321 | 5.50 |
| 2006 | 63 498 | 214 | 3.37 | 294 | 4.63 | 366 | 5.76 |
| 2007 | 77 658 | 206 | 2.65 | 302 | 3.89 | 358 | 4.61 |
| 2008 | 80 981 | 199 | 2.46 | 301 | 3.72 | 357 | 4.41 |
| 2009 | 89 353 | 221 | 2.47 | 312 | 3.49 | 369 | 4.13 |
| 2010 | 90 583 | 187 | 2.06 | 298 | 3.29 | 377 | 4.16 |
| 2011 | 1 10 033 | 207 | 1.88 | 313 | 2.84 | 375 | 3.41 |
| 2012 | 1 32 227 | 253 | 1.91 | 379 | 2.87 | 435 | 3.29 |
| 2013 | 1 27 015 | 192 | 1.51 | 295 | 2.32 | 366 | 2.88 |
| 2014 | 1 52 929 | 229 | 1.50 | 356 | 2.33 | 442 | 2.89 |
| 2015 | 1 26 648 | 200 | 1.58 | 315 | 2.49 | 391 | 3.09 |
| AARR (%) | | −7.67 | | −6.76 | | −6.79 | |
| $\chi^2$ | | 3200.17 | | 3029.94 | | 4514.08 | |
| p Value | | <0.01 | | <0.01 | | <0.01 | |

U5MR, under-5 mortality rate defined as the number of deaths within 5 years from birth per 1000 live births.
NMR, neonatal mortality rate defined as the number of deaths within 28 days from birth per 1000 live births.
IMR, infant mortality rate defined as the number of deaths within 1 year from birth per 1000 live births.
AARR, annual average rate of reduction.

five leading causes of death were pneumonia (19.01%), birth asphyxia (13.15%), congenital heart disease (11.66%), preterm/low birth weight (8.53%) and other congenital abnormalities (7.75%) in 1992 in that order whereas, in 2015, this order became preterm/low birth weight (18.41%), congenital heart disease (12.53%), birth asphyxia (10.23%), other congenital abnormalities (8.95%) and pneumonia (7.42%). In particular, pneumonia dropped from first place (19.01%) in 1992 to fifth place (6.78%) in 1998 whereas congenital heart disease climbed from third place (11.66%) to first place (14.73%) during the same period, and remained the leading cause of death for 12 consecutive years from 1998 to 2009. Furthermore, pneumonia was not among the five leading causes of death during 1999, 2003–2005

and 2007–2009. However, other neonatal diseases climbed to fifth place within the same period.

The number of deaths due to pneumonia, birth asphyxia, congenital heart disease, preterm/low birth weight and other congenital abnormalities declined by 92.26%, 84.56%, 78.68%, 57.18% and 77.08%. Among the five leading causes of death during this time period, the fastest rate of decline was observed in pneumonia, the AARR of which was 10.53%; and the slowest rate of decline was observed in preterm/low birth weight, the AARR of which was only 3.62%. In particular, preterm/low birth weight showed a steady overall increase of 44.42% from 2013 to 2015. A similar trend, however, was not evident in other diseases. With respect to all causes of death, deaths due to neural tube defects showed a decline

**Table 2** Mortality rates of the leading causes of death in children under 5 years old in Beijing, China, 1992–2015

| Year | Pneumonia Mortality rate (%) | Rank order | Birth asphyxia Mortality rate (%) | Rank order | Congenital heart disease Mortality rate (%) | Rank order | Preterm/low birth weight Mortality rate (%) | Rank order | Other congenital abnormalities Mortality rate (%) | Rank order | Other neonatal diseases Mortality rate (%) | Rank order |
|---|---|---|---|---|---|---|---|---|---|---|---|---|
| 1992 | 295.96 (19.01) | 1 | 204.62 (13.15) | 2 | 181.47 (11.66) | 3 | 132.76 (8.53) | 4 | 120.58 (7.75) | 5 | 76.73 (4.93) | 6 |
| 1993 | 250.83 (16.79) | 1 | 175.44 (11.74) | 2 | 168.59 (11.28) | 3 | 138.43 (9.27) | 4 | 134.32 (8.99) | 5 | 87.72 (5.87) | 6 |
| 1994 | 213.12 (16.68) | 1 | 140.25 (10.98) | 4 | 149.87 (11.73) | 2 | 141.62 (11.09) | 3 | 125.12 (9.80) | 5 | 82.50 (6.46) | 6 |
| 1995 | 184.38 (14.90) | 1 | 168.90 (13.65) | 2 | 153.42 (12.40) | 3 | 125.27 (10.13) | 5 | 129.49 (10.47) | 4 | 56.30 (4.55) | 7 |
| 1996 | 144.76 (12.53) | 2 | 166.10 (14.38) | 1 | 143.24 (12.40) | 3 | 109.71 (9.50) | 5 | 121.90 (10.55) | 4 | 92.95 (8.05) | 6 |
| 1997 | 113.24 (10.25) | 4 | 187.23 (16.94) | 1 | 144.95 (13.11) | 2 | 107.20 (9.70) | 5 | 129.85 (11.75) | 3 | 61.91 (5.60) | 6 |
| 1998 | 65.51 (6.78) | 5 | 131.01 (13.57) | 2 | 142.24 (14.73) | 1 | 95.45 (9.88) | 4 | 123.53 (12.79) | 3 | 54.28 (5.62) | 6 |
| 1999 | 73.41 (7.74) | 6 | 116.71 (12.30) | 3 | 171.30 (18.06) | 1 | 88.47 (9.33) | 4 | 122.36 (12.90) | 2 | 79.06 (8.33) | 5 |
| 2000 | 51.36 (6.62) | 5 | 117.40 (15.13) | 2 | 170.60 (21.99) | 1 | 56.87 (7.33) | 4 | 95.39 (12.29) | 3 | 38.52 (4.96) | 6 |
| 2001 | 51.61 (6.63) | 5 | 109.41 (14.06) | 2 | 115.60 (14.85) | 1 | 94.96 (12.20) | 3 | 88.76 (11.41) | 4 | 47.48 (6.10) | 6 |
| 2002 | 52.59 (7.41) | 5 | 97.08 (13.68) | 2 | 125.40 (17.66) | 1 | 86.97 (12.25) | 3 | 84.95 (11.97) | 4 | 42.47 (5.98) | 6 |
| 2003 | 31.25 (3.96) | 6 | 109.38 (13.68) | 2 | 140.63 (17.82) | 1 | 85.94 (10.89) | 3 | 70.31 (8.91) | 4 | 62.50 (7.92) | 5 |
| 2004 | 24.68 (4.51) | 6 | 65.06 (11.89) | 4 | 103.20 (18.85) | 1 | 87.50 (15.98) | 2 | 67.31 (12.30) | 3 | 47.11 (8.61) | 5 |
| 2005 | 18.83 (3.43) | 7 | 78.75 (14.33) | 2 | 90.74 (16.51) | 1 | 68.48 (12.46) | 4 | 71.91 (13.08) | 3 | 51.36 (9.35) | 5 |
| 2006 | 42.52 (7.38) | 5 | 72.44 (12.57) | 2 | 107.09 (18.58) | 1 | 58.27 (10.11) | 4 | 66.14 (11.48) | 3 | 33.07 (5.74) | 6 |
| 2007 | 30.90 (6.70) | 6 | 70.82 (15.36) | 2 | 84.99 (18.44) | 1 | 45.07 (9.78) | 4 | 47.64 (10.34) | 3 | 33.48 (7.26) | 5 |
| 2008 | 27.17 (6.16) | 6 | 56.80 (12.89) | 2 | 64.21 (14.57) | 1 | 38.28 (8.68) | 4 | 51.86 (11.76) | 3 | 34.58 (7.84) | 5 |
| 2009 | 26.86 (6.50) | 6 | 53.72 (13.01) | 2 | 73.86 (17.89) | 1 | 34.69 (8.40) | 4 | 36.93 (8.94) | 3 | 27.98 (6.78) | 5 |
| 2010 | 33.12 (7.96) | 5 | 38.64 (9.28) | 3 | 56.30 (13.53) | 2 | 61.82 (14.85) | 1 | 34.22 (8.22) | 4 | 14.35 (3.45) | 9 |
| 2011 | 34.54 (10.13) | 4 | 32.72 (9.60) | 5 | 41.81 (12.27) | 2 | 40.90 (12.00) | 3 | 53.62 (15.73) | 1 | 18.18 (5.33) | 6 |
| 2012 | 22.69 (6.90) | 5 | 55.21 (16.78) | 1 | 38.57 (11.72) | 3 | 48.40 (14.71) | 2 | 34.79 (10.57) | 4 | 19.66 (5.98) | 6 |
| 2013 | 22.04 (7.65) | 5 | 35.43 (12.30) | 2 | 34.64 (12.02) | 3 | 39.37 (13.66) | 1 | 33.85 (11.75) | 4 | 11.81 (4.10) | 8 |
| 2014 | 16.35 (5.66) | 5 | 26.81 (9.28) | 4 | 39.23 (13.57) | 2 | 54.27 (18.78) | 1 | 34.66 (11.99) | 3 | 5.23 (1.81) | 14 |
| 2015 | 22.90 (7.42) | 5 | 31.58 (10.23) | 3 | 38.69 (12.53) | 2 | 56.85 (18.41) | 1 | 27.64 (8.95) | 4 | 6.32 (2.05) | 13 |
| AARR(%) | −10.53 | | −7.80 | | −6.50 | | −3.62 | | −6.20 | | −10.28 | |

Mortality rates are presented as deaths per 100 000 live births per year.
AARR, annual average rate of reduction.

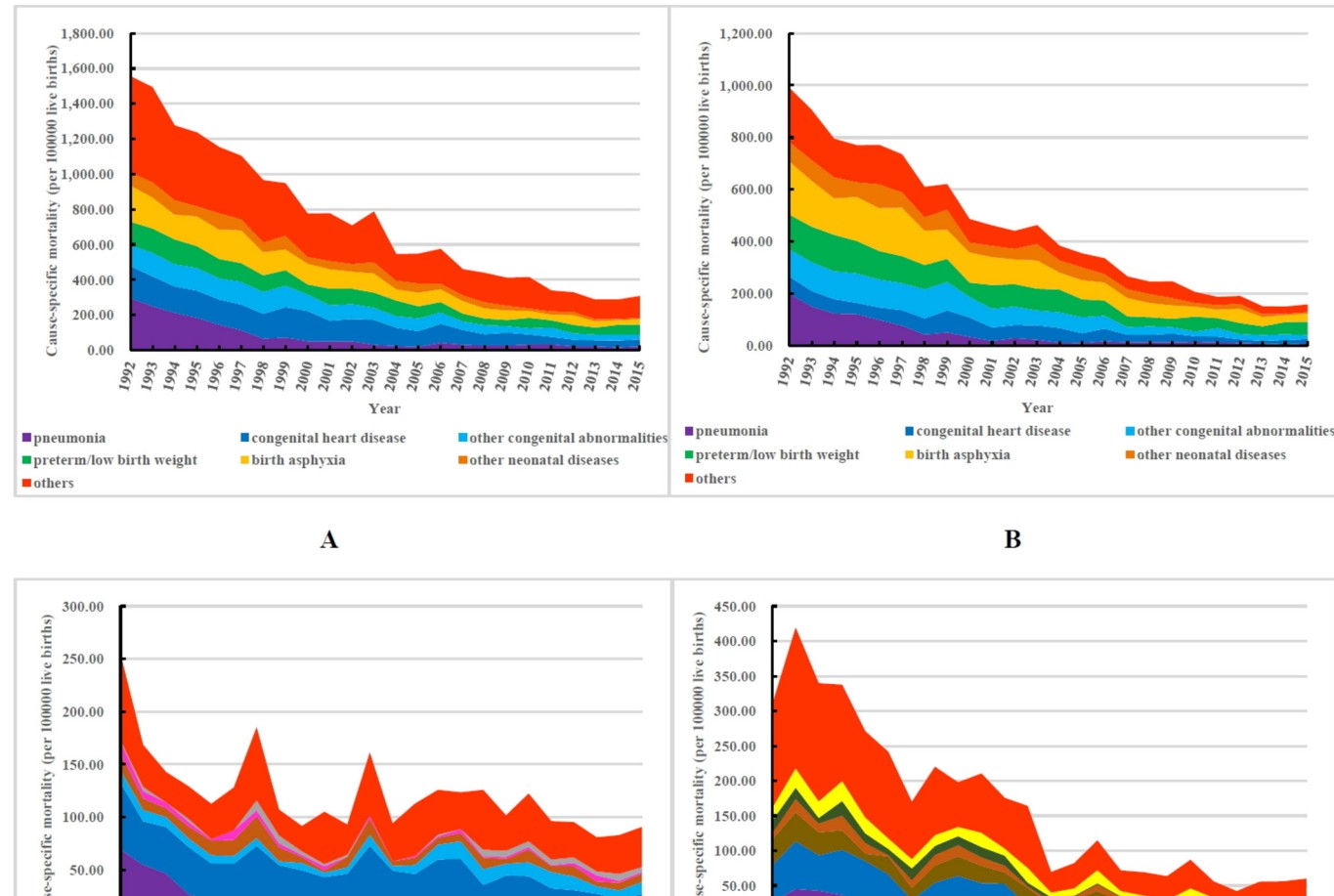

**Figure 2** Trends in cause-specific mortality rates among children of different ages in Beijing, China, 1992–2015: (A) children under 5 yers old; (B) neonates (<28 days); (C) 28 days to <1 year; (D) 1–4 years.

by 98.94%, which was the greatest reduction observed; however, deaths due to endocrine, nutritional and metabolic diseases showed an increasing trend by as much as 110.70%.

Figure 2 illustrates the trends in cause-specific mortality rates among different age groups (children aged <28 days; 28 days to 1 year; and 1–4 years) from 1992 to 2015. The overall trend of decline in mortality rates and the causes of death in under-5 children were similar to those observed in neonates (<28 days) (figure 2 A and B) but different from those among the age groups 28 days to 1 year and 1–4 years, which exhibited slightly increasing peaks in 1998, 2003 and 2010, and 1993, 1999, 2006 and 2010, respectively (figure 2 C and D). The main causes of death among children aged 28 days to 1 year were accidental asphyxia, sepsis and other neurological disease, in addition to pneumonia, congenital heart disease and other congenital abnormalities. Furthermore, the cause-specific mortality rates due to congenital heart disease, other

congenital abnormalities and accidental asphyxia showed obvious fluctuations with the largest annual variations of 77.08% (from 31.71 in 1997 to 56.15 in 1998 per 100 000 live births), 94.90% (from 3.76 in 1999 to 7.34 in 2000 per 100 000 live births) and 389.90% (from 2.06 in 2001 to 10.11 in 2002 per 100 000 live births) (figure 2C). The main causes of death in children aged 1–4 years were tumours (leukaemia and other tumours) and accidents (drowning, traffic accident, accidental asphyxia, accidental poisoning, accidental fall and other accidents) in addition to pneumonia and congenital heart disease. The cause-specific mortality in these diseases also showed significant fluctuations, but were all reduced by varying degrees after 1992 (figure 2D).

## Forecasting model for the U5MR in Beijing

We used the U5MRs from 1992 to 2013 as the training sample to fit the forecasting model through stabilisation, model identification, parameter estimation and model

**Table 3** Autocorrelation and partial autocorrelation functions of the first difference time series

| Autocorrelation | Partial correlation | | AC | PAC | Q-Stat | p Value |
|---|---|---|---|---|---|---|
| | | 1 | −0.448 | −0.448 | 4.840 | 0.028 |
| | | 2 | 0.193 | −0.009 | 5.788 | 0.055 |
| | | 3 | −0.094 | −0.014 | 6.027 | 0.110 |
| | | 4 | 0.336 | 0.363 | 9.239 | 0.055 |
| | | 5 | −0.235 | 0.068 | 10.913 | 0.053 |
| | | 6 | 0.151 | 0.023 | 11.651 | 0.070 |
| | | 7 | 0.070 | 0.170 | 11.821 | 0.107 |
| | | 8 | −0.105 | −0.158 | 12.231 | 0.141 |
| | | 9 | −0.129 | −0.306 | 12.897 | 0.167 |
| | | 10 | 0.154 | −0.082 | 13.932 | 0.176 |
| | | 11 | −0.011 | 0.032 | 13.938 | 0.236 |
| | | 12 | −0.199 | −0.106 | 16.060 | 0.189 |
| | | 13 | 0.080 | 0.031 | 16.449 | 0.226 |
| | | 14 | −0.079 | −0.093 | 16.879 | 0.263 |
| | | 15 | −0.036 | −0.065 | 16.985 | 0.320 |
| | | 16 | −0.088 | −0.016 | 17.733 | 0.340 |
| | | 17 | 0.021 | −0.181 | 17.788 | 0.402 |
| | | 18 | −0.056 | −0.029 | 18.300 | 0.436 |
| | | 19 | −0.025 | 0.106 | 18.451 | 0.493 |
| | | 20 | 0.000 | 0.032 | 18.451 | 0.558 |

AC, autocorrelation; PAC, partial correlation; Q-Stat, Q-statistic.

diagnosis. The U5MRs in Beijing from 1992 to 2013 showed a non-stationary declining trend with time. After the first notable difference, the sequence chart showed no obvious changes in trend, as confirmed by the unit root test ($t$=−3.05, p<0.05). Therefore, the difference in order (d) was 1. From table 3, the autocorrelation function and partial autocorrelation function charts both showed trailing and declined to 0 after lag 1. Thus, we confirmed that p=1 and $q$=1. The model was confirmed to be ARIMA (1,1,1). The model coefficients were not significantly zero according to the $t$-test (p<0.05, $R^2$=0.982). The Akaike information criterion and Bayes' information criterion values of the ARIMA (1,1,1) model were 2.15 and 2.30, respectively. The results of parameter estimation were incorporated with the general formula of the model to obtain the forecasting model for the under-5 mortality rate as follows:

$$x_t = -0.445 + 1.509x_{t-1} - 0.509x_{t-2} + \alpha_t + 0.999\alpha_{t-1}$$

The autocorrelation and partial autocorrelation values of the residual plots of the ARIMA (1,1,1) model were all within the 95% CI (p>0.05) based on the Box–Ljung test. Therefore, the assumption that the overall autocorrelation function was zero stood and met the criteria for white noise. Thus, the forecasting model we established was appropriate.

The surveillance data from 1992 to 2013 and 2014–2015 was used for internal and external validation of the model. Table 4 and figure 3 show the time series of the predictive U5MR based on the forecasting model. The actual values all fell within the 95% CI and the mean absolute percentage error was 4.76%. Furthermore, statistically significant differences between the predictive and actual values were not observed in either the internal or external validation, based on the results of the Wilcoxon test ($Z_{internal}$ = −0.840, $P_{internal}$ = 0.401 > 0.05; $Z_{external}$ = −0.447, $P_{external}$ = 0.655 > 0.05). Therefore, the internal and external prediction effects of the forecasting model were both favourable.

The predictive value of the U5MRs from 2016 to 2020 were 2.88‰, 2.87‰, 2.90‰, 2.97‰ and 3.09‰ based on the forecasting model, which indicated a slight upward trend.

## DISCUSSION
The mortality rates in neonates, infants and under-5 children from 1992 to 2015 in Beijing, China all showed an overall trend of decline, with reductions by 84.06%, 80.04% and 80.17%, respectively. Neonates, in particular, had the highest AARR of 7.67%. In 2011, the mortality rates in infants and under-5 children was reduced to 2.84‰ and 3.41‰, respectively, thereby achieving one of the aims (to reduce infant mortality to <4‰ and the U5MR to <5‰) of the Beijing Twelfth Five-year Period Children Development Plan (2011–2015) 4years ahead

**Table 4** Results of validation in the ARIMA (1, 1, 1) model

| Year | Actual values (‰) | Predictive values (‰) | 95% CI | | Relative error (%) |
| --- | --- | --- | --- | --- | --- |
| | | | Upper | Lower | |
| 1992 | 15.57 | – | – | – | – |
| 1993 | 14.94 | 14.58 | 12.91 | 16.25 | –2.41 |
| 1994 | 12.77 | 13.76 | 12.50 | 15.03 | 7.75 |
| 1995 | 12.37 | 12.84 | 11.68 | 14.00 | 3.80 |
| 1996 | 11.55 | 11.69 | 10.57 | 12.80 | 1.21 |
| 1997 | 11.05 | 10.82 | 9.73 | 11.91 | –2.08 |
| 1998 | 9.66 | 9.98 | 8.91 | 11.05 | 3.31 |
| 1999 | 9.49 | 9.37 | 8.32 | 10.43 | –1.26 |
| 2000 | 7.76 | 8.51 | 7.47 | 9.56 | 9.66 |
| 2001 | 7.78 | 8.08 | 7.04 | 9.12 | 3.86 |
| 2002 | 7.10 | 7.22 | 6.18 | 8.25 | 1.69 |
| 2003 | 7.89 | 6.65 | 5.62 | 7.68 | –15.72 |
| 2004 | 5.47 | 5.80 | 4.77 | 6.82 | 6.03 |
| 2005 | 5.50 | 5.85 | 4.83 | 6.87 | 6.36 |
| 2006 | 5.76 | 5.21 | 4.19 | 6.22 | –9.55 |
| 2007 | 4.61 | 4.61 | 3.59 | 5.63 | 0 |
| 2008 | 4.41 | 4.46 | 3.44 | 5.47 | 1.13 |
| 2009 | 4.13 | 4.07 | 3.05 | 5.08 | –1.45 |
| 2010 | 4.16 | 3.76 | 2.75 | 4.77 | –9.62 |
| 2011 | 3.41 | 3.43 | 2.42 | 4.44 | 0.59 |
| 2012 | 3.29 | 3.37 | 2.36 | 4.38 | 2.43 |
| 2013 | 2.88 | 3.17 | 2.16 | 4.17 | 10.07 |
| 2014 | 2.89 | 3.10 | 2.09 | 4.10 | 7.27 |
| 2015 | 3.09 | 2.90 | 1.86 | 3.94 | –3.97 |

of schedule. In 2015 the U5MR was further reduced to 3.09‰, the lowest in China,[11–13] which was close to the levels achieved in developed countries such as Japan (2.70‰) and Singapore (2.70‰).[14 15] Thus, Beijing has made considerable progress in reducing mortality of under-5 children over the past 24 years. Nevertheless, the mortality rates in neonates, infants and under-5 children all increased from 2002 to 2003 and from 2014 to 2015.

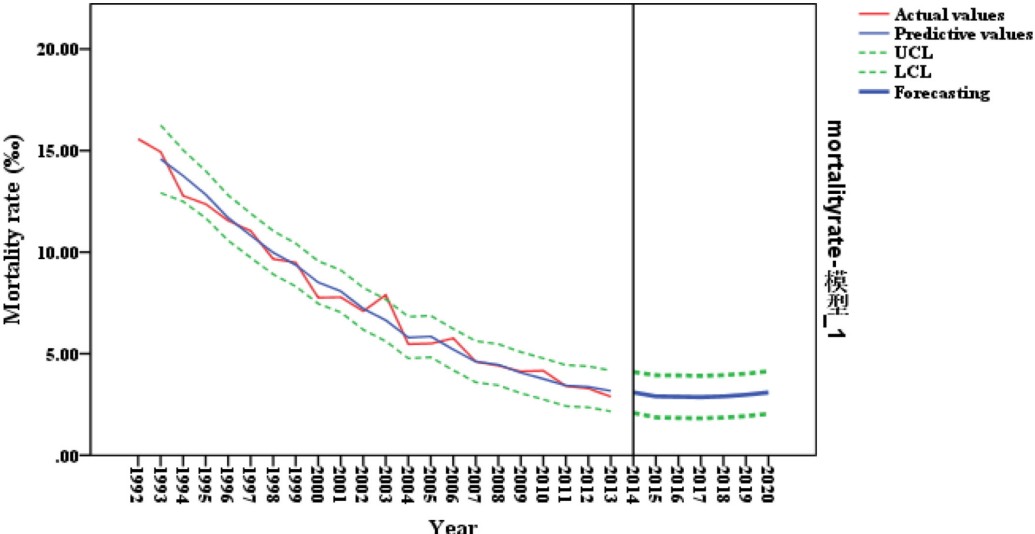

**Figure 3** Prediction of ARIMA (1,1,1) model. UCL, upper control limit; LCL, lower control limit.

The increase in 2003 was related to the severe acute respiratory syndrome (SARS) outbreak in Beijing, whereas the increase in 2015 might have been related to the Selective Two-child Policy regulation issued in the previous year.

Several studies have reported an increase in the mean maternal age at childbirth in Beijing since the Selective Two-child Policy was implemented in 2014.[10 16] Advanced maternal age has been a proven risk factor of many disorders in under-5 children, such as prematurity and congenital abnormalities.[17–19] In the present study, the mortality rates in neonates, infants and under-5 children increased slightly in 2015, by 5.33%, 6.87% and 6.92% compared with 2014. Furthermore, the cause-specific mortality rate of preterm/low birth weight increased by 44.42% during the period 2013–2015. However, the cause-specific mortality rate of other diseases fluctuated with regular changes in interval. Therefore, we suspect that the Selective Two-child Policy may induce only a slight upward trend in the U5MR in Beijing instead of a major boost. This finding could be attributed mainly to the high level of healthcare and medical treatment available and is less likely caused by the influence of the policy on the U5MR. Using the forecasting model established in the present study, we predicted the U5MRs from 2016 to 2020 to be 2.88‰, 2.87‰, 2.90‰, 2.97‰ and 3.09‰, respectively, indicating a slight upward trend over the next 5 years. Thus, our suggestion is that maternal and child healthcare in Beijing still faces challenges and greater investment and attention should be provided to health facilities involved in this field.

Deaths due to birth asphyxia decreased by 84.56% from 1992 to 2015 in Beijing. On the one hand, it benefited from intensification of intrapartum monitoring and promotion of neonatal recovery technology in Beijing while, on the other hand, it may be related to the increase of caesarean sections. There have been several studies verifying that the increase in the caesarean section rate can reduce the occurrence of birth asphyxia.[20 21] There has been a sharp rise in the caesarean section rate in Beijing since 1999, which reached 57% during 2009 and 2013,[22] which may play a part in reducing the deaths due to birth asphyxia, whereas birth asphyxia remained the first and second causes of death in 2012 and 2013, respectively. Increased maternal age, obesity and diabetes are the risk factors for birth asphyxia,[23–25] suggesting that maternal education programmes targeted at high-risk groups will probably further reduce the number of deaths due to birth asphyxia.

Congenital heart disease has been the leading cause of death in under-5 children in Beijing since 1998. In 2009, Beijing established a local children's congenital heart disease screening and referral network. With the aim of achieving early detection, diagnosis and treatment, the network has improved the survival rate and quality of life for children with congenital heart disease. In 2010, congenital heart disease dropped to become the second leading cause of death. However, the cause-specific mortality of congenital heart disease in children

aged 28 days to 1 year and 1–4 years showed significant fluctuations. These values increased during some years, thus preventing the U5MR in Beijing from declining further. The maternal age in Beijing is increasing yearly and elderly parturient accounted for 20.04% in 2014.[10] Furthermore, the increased maternal age is closely related to the occurrence of gestational hypertension and gestational diabetes, all of which are risk factors for congenital heart disease.[18 26] Therefore, we should advocate pregnancy at an appropriate age and improve prenatal diagnosis to decrease the occurrence of congenital heart disease.

Preterm/low birth weight continued to be one of the five leading causes of death in under-5 children in Beijing, which may be explained by the increasing rate of caesarean sections. Older parturients and multiple births by assisted reproductive techniques both promoted the rate of caesarean section, which reached 57% in 2014 in Beijing.[22] Special prenatal care services might need to be provided to pregnant women of an advanced age and those becoming pregnant using assisted reproductive technologies. Recent research suggests that higher levels of particulate matter of diameter <2.5 μm ($PM_{2.5}$) are associated with a higher risk of low birth weight.[27] Therefore, given the levels of severe air pollution in Beijing, this should be a major point of concern to reduce the incidence of low birth weight.

Pneumonia showed the greatest and fastest rate of reduction among the five leading causes of death, and this contributed considerably to the reduced U5MR in Beijing. This might be related to the implementation of neonatal visits and the prevention of childhood respiratory infections.[28] The remarkable reduction in mortality caused by neural tube defects was probably due to the increased intake of folic acid supplements among pregnant women in Beijing (One Hundred Achievements Promotion Plan in Ten Years in China). However, deaths due to endocrine, nutritional and metabolic disease showed a general trend of increase by 110.70%, from 1992 to 2015. This might be related to older parturients and poor dietary practices; however, further studies and effective preventative measures are still required.

It is worth noting that trends in mortality rates and the composition of the causes of death were significantly different in children of different ages. Neonatal mortality (<28 days) accounted for more than 50% of deaths in under-5 children. Therefore, the general trends in mortality and specific of causes of death were significantly influenced by neonates. Nevertheless, the conditions in children aged 28 days to 1 year and 1–4 years showed obvious differences. Therefore, different aspects should receive greater attention among the various age groups, so that different preventative measures can be adopted for children of different ages. For children aged 28 days to 1 year, the key focus of preventative measures should be accidental asphyxia through training of the guardians in the relevant knowledge and skills. For children aged 1–4 years, the key focus of preventative measures should be

accidental death. The 'Safe Beijing' action plan launched in 2006[29] is likely to effectively reduce the number of accidents in children.

Our study has several advantages. First, the data are reliable because our study considers the entire population, based on the Beijing Under-5 Mortality Rate Surveillance Network and Beijing Maternal and Child Healthcare System which covers all under-5 children in all districts of Beijing. Thus, the present study is not limited by a small sample size or low coverage rate. Second, our study revealed the long-term trends in mortality rates and leading causes of death in under-5 children through a 24-year analysis. Furthermore, we reported the health conditions of under-5 children following implementation of the Selective Two-child Policy in Beijing in 2014. The results of our study will provide suggestions for the government to assess the present policy and formulate further regulations related to maternal and child healthcare. However, several limitations should be considered when interpreting our findings. First, because of the unique regional divisions in Beijing and the lack of specific data on each district, we did not analyse the differences between rural and urban areas. Second, the ARIMA (1,1,1) model forecasts future U5MRs reliably, but only for a short period. This is because the model only considered variations in mortality with time, instead of other possible impacting factors such as environment, economy and politics. Data must be continually updated to predict further mortality rates. Other models which can predict U5MRs more accurately and capture its patterns more specifically are needed. Thirdly, since we only collected 2-year data (2014 and 2015) after the Selective Two Child Policy (2014), the representativeness of the changing trend and solid statistical analysis is limited. The relation between the changing trend in U5MRs and policy regulation is still unclear and is needed to be observed for further years. We could only give a suggestion to raise healthcare workers' awareness about this potential problem.

## CONCLUSIONS

From 1992 to 2015, Beijing has made considerable progress in reducing the mortality rates of under-5 children. However, the U5MR might show a slight upward trend from 2016 to 2020. The key points for future consideration regarding child healthcare include the management of birth asphyxia, congenital heart disease, preterm/low birth weight and other congenital abnormalities. To further reduce the U5MR in Beijing, specific prevention measures should be adopted for children of various age groups.

**Author affiliations**
[1]Department of Epidemiology and Health Statistics, School of Public Health, Capital Medical University, Beijing, China
[2]Beijing Key Laboratory of Clinical Epidemiology, Beijing, China
[3]Department of Children's Health Care, Beijing Obstetrics and Gynecology Hospital, Capital Medical University, Beijing, China
[4]Department of Child, Adolescent Health and Maternal Health, School of Public Health, Capital Medical University, Beijing, China
[5]Department of Hospital Management, School of Health Administration and Education, Capital Medical University, Beijing, China

**Acknowledgements** We thank all healthcare workers in the surveillance system for their efforts in data collection.

**Contributors** HC: Study design, database establishment, statistical analysis and writing paper. JW: Study design, collecting data, database establishment and writing paper. YL and DL: Study design and collecting data. JG, YH, KM, DH, BL, ZL, HQ: Database establishment. LZ: study design, direction of statistical analysis and writing and modification of the paper.

**Funding** This work was supported by National Key R&D Program of China grant number 2016YFC0900603.

**Competing interests** None declared.

**Patient consent** Not obtained as the data used in this study were based on Children's Death Reporting Card from the Beijing Under-5 Mortality Rate Surveillance Network.

**Provenance and peer review** Not commissioned; externally peer reviewed.

**Data sharing statement** All health workers in the Beijing Under-5 Mortality Rate Surveillance Network and people who get eligibility of application can use the data.

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
