## [Reviewer comments · BMJ Open]

ARTICLE DETAILS

TITLE (PROVISIONAL)	Trend analysis of mortality rates and causes of death in under-5 children in Beijing, China from 1992 to 2015 and forecast of mortality into the future: an entire population-based epidemiological study
AUTHORS	Cao, Han; Wang, Jing; Li, Yichen; Li, Dongyang; Guo, Jin; Hu, Yifei; Meng, Kai; He, Dian; Liu, Bin; Liu, Zheng; Qi, Han; Zhang, Ling

VERSION 1 - REVIEW

REVIEWER	Juhwan Oh Seoul National University College of Medicine, Rep of Korea
REVIEW RETURNED	21-Feb-2017

GENERAL COMMENTS	The paper discloses interesting findings in terms of secular trends in mortality and major cause of death in Beijing. However, the manuscript has some point to be revised to appropriately show its value to the audience. Mortality in 2015 could not be interpreted as the results of “Selective Two child policy” implemented in 2014. Furthermore, the alleged reason (the aged mother) is only attributable for neonatal mortality not for infant and U5 Mortality in this mechanism. However, increasing in neonatal mortality (+0.08) is relatively smaller than the other two mortalities (+0.16, +0.20). Removing the “Selective Two child policy” attribution is suggested. If this should be the main research question, the method should be more solid rather than this secular trend analysis. Otherwise, the authors need to take out this research question. (in results section). Page 7 line 15-19. It is a bit confusing in the text. U5 death included Infant death or not? Similarly, infant death include neonate death or not? According to the text, it looks that they are not included to the others (mutually exclusive). However, next sentence “death in neonates accounted for more than 50% of all death among for under-5 children” made audience to think they not mutually exclusive to each other (U5 death include infant death; and infant death include neonate death). Which one is correct? Please be consistent or make it not confusing. New row for total death next to 2015 will make audience less confusing as well. Page 8. Line 38. “mortality rates for leading cause of death...overall decreasing...” sounds very illogical. Some mortality rates, which were leading cause of death in 1992, can be decreasing since then. However, general decreasing of leading cause of deaths sounds very inappropriate expression. Other cause became major leading causes? Or Same rank of cause of death but all decreasing in terms of mortality per se? Even the latter is right, this section is not the
--

	decomposition of all cause mortality, but cause of death distribution mostly based on rank order description. So in this sense, the expression is not suitable to the section heading. Page 10, line 3. The percentages looks confusing? What do they mean? Upward trend forecasting may be model misspecification due to adapting the function of “$y=aX^2+bx+c$”. Inverse root X function or logarithmic function would predict differently. Page 14. Line 34-42. It sounds too redundant. Fig 1. Same color should be consistent across A-D so that no confusion rises to the audience. Purple represented pneumonia in A-D consistently. However, yellow color was for birth asphyxia in A-B but in C, it meant septicemia; in D, it meant leukemia. By this way, chart looks not succinctly but the mortality composition change but making confusion. Please use different color different cause of death.
--	--

REVIEWER	Melkamu Dedefo University of KwaZulu-Natal, South Africa Haramaya University, Ethiopia Kersa HDSS, Ethiopia
REVIEW RETURNED	24-Feb-2017

GENERAL COMMENTS	General comment: Trend analysis in mortality rates and causes of death for under-5 children in Beijing, China from 1992 to 2015 and forecasting mortality into the future: an entire population based epidemiological study. The topic is an important one and covers a timely and important subject of great medical interest: Future death rates are extremely important to governmental and non-governmental organizations as forecasts of mortality rates used to plan social security and health care programs. The paper is well written and the conclusions drawn more or less supported by the data used. The limitations of the study were stated fairly though the strength was overstated. Latest advances in predicting mortality increasingly include Bayesian inference and they account for the distribution of the age at death in order to capture detailed patterns of mortality. In addition, some recent studies used rates of improvement or rate changes rather than death rates to forecast mortality changes better. Whether you use classical or Bayesian approach, to address the objectives in your paper more efficiently:  1) It would be better to use forecasting models which accounts for different age groups (infants, neonates,...). 2) It would be better to forecast mortality for each major cause of death. 3) It would be better to use forecasting models which accounts for other related impact factors. Additional comments:
--

	A. Introduction This part is described and organized well. 1) Line 13-14 it would be better to rephrase the statement for better meaning. B. Methods 2) Line 44-46 it would be better to rewrite the statement for better meaning 3) Line 53-56 check the sentence for coherence or may be missing statement. C. Results 4) On page 7 line 17 (death in neonates accounted for more than 50% of all deaths...) not coincides with the figure in table 1. 5) Page 8 line 21-26 it would be better to rephrase for better meaning. 6) Page 8 line 38-42 (... Decreased by ...) it is not clear for which time period or relative to what the figure stands for. 7) Page 8 line 44-49 check for coherence or sentence fragment. 8) On page 9 table 2, it would be more informative to put mortality rate for all years. There will be a possibility of higher mortality rate even for those having rank order more than five. 9) Page 10 line 9-11 it would be better to rephrase the sentence for better meaning. 10) Page 43-48 Discussion 11) Page 12 line 49 better to rewrite the statement 12) Page 13 line 13-14 sentence fragment, formal writing 13) Page 13 line 39-40 check for sentence fragment. 14) Page 13 line 47-52 it would be better to rephrase the sentence for better meaning. 15) Page 14 line 28 rewrite the statement. 16) Page 14 39-43 check for sentence coherence. 17) Page 15 line 2-28 As I go through your paper unable to find a model incorporating the factor "Selective Two-child Policy". I wonder if you could justify how you come to this decision. ABSTRACT Method 1) Line 14 (...analyzed using statistical description...) It would be better to rephrase the statement for better meaning. Result 2) Line 21 sentence defragment. 3) Line 21-31 inconsistencies such as: to vs - and sentence fragment. Conclusion 4) Line 36 (...U5MRs may show a slightly upward trend from 2016 to 2020 as influenced by the "Selective Two-child Policy"...). As I go through your paper unable to find a model incorporating the factor "Selective Two-child Policy". I wonder if you could justify how you come to this decision.
--	---

REVIEWER	Peige Song Centre for Global Health Research, Usher Institute of Population Health Sciences and Informatics, The University of Edinburgh UK
REVIEW RETURNED	14-Mar-2017

GENERAL COMMENTS	1) The abstract needs to be rephrased. For example, page 2 line 21: please specify the corresponding periods for the three decreasing rates; Several syntax errors;
---

	2) Page 4 line 8, line 11, line 19, please add references; 3) Ref 7, needs to be updated by: He, Chunhua, et al. "National and subnational all-cause and cause-specific child mortality in China, 1996–2015: a systematic analysis with implications for the Sustainable Development Goals." The Lancet Global Health (2016). Song, Peige, et al. "Causes of death in children younger than five years in China in 2015: an updated analysis." Journal of global health 6.2 (2016). 4) Please change the glossary and specific diseases and to be in accordance with the abovementioned two national studies; 5) The writing should be improved with assistance of English native speakers; 6) Data between 1992 and 2002 was collected using “double entry and consistency check”, is it the same for the “electronic cards” between 2003 and 2015? Please clarify; 7) Reference the “MCMS Child death report card” and all disease definitions in the Online Supplementary Document of “Causes of death in children younger than five years in China in 2015: an updated analysis” (https://www.ncbi.nlm.nih.gov/pmc/articles/PMC5140075/#S1) 8) Table 1, clarify the unit of “Mortality rate”; 9) Table 2, add the proportions of death causes, not only the rank orders; And rephrased the corresponding results accordingly; 10) Page12, line 39-32 “Deaths due to birth asphyxia declined steadily through Beijing intensifying intrapartum monitoring and promotion of neonatal recovery technology”, please discuss if caesarean section plays a role in the decline of deaths due to birth asphyxia; 11) Page12, line 51-52: “Several studies have verified that congenital heart disease has several influence factors, such as increased maternal age, prenatal infections, gestational hypertension, gestational diabetes, etc.” This kind of discussion is hand-waving, further in-depth discussion is needed; 12) Page12, line 58; page 13 line 4-11: please discuss if caesarean section plays a role; 13) Throughout the discussion part, DO NOT only cite other researchers’ statements as explanations of changes of death causes, explore possible reasons in BEIJING-CONTEXT, this is of more interest for readers, and more useful for policy-makers;
--	--

VERSION 1 – AUTHOR RESPONSE

Reviewer: 1

1. Mortality in 2015 could not be interpreted as the results of “Selective Two child policy” implemented in 2014. Furthermore, the alleged reason (the aged mother) is only attributable for neonatal mortality not for infant and U5 Mortality in this mechanism. However, increasing in neonatal mortality (+0.08) is relatively smaller than the other two mortalities (+0.16, +0.20). Removing the “Selective Two child policy” attribution is suggested. If this should be the main research question, the method should be more solid rather than this secular trend analysis. Otherwise, the authors need to take out this research question.

Reply: Thanks for your comments.

We agree with reviewer’s comments. Since we only collected 2-year data (2014 and 2015) after the “Selective Two Child Policy” (2014), the representativeness of the changing trend and solid statistical analysis was limited. Therefore, our results could not detect the effect of “Selective Two Child Policy” directly. We just suggest that the policy regulation may affect slight upward trend in mortalities to rise

healthcare workers' attention about this potential problem (as we discussing in "Limitations", Page 3, Line 17-20 and Page 15, Line 15-19).

It seems we haven't explained clearly the three mortalities, so we clarify the definition of "neonatal mortality rate (NMR)", "infant mortality rate (IMR)" and "under-5 mortality rate (U5MR)" in methods section (Page 6, Line 2-6). That is to say, U5MR included IMR, and IMR included NMR. Therefore, increasing in NMR (+0.08) must be smaller than IMR (+0.16) and U5MR (+0.20). However, increasing in mortalities in neonates (<28 days) and children aged 28 days to 1 year were bigger than children aged 1-4 years indeed after we calculated mortalities in different age groups. Increasing in mortalities in neonates and children aged 28 days to 1 year and 1-4 years were 0.08, 0.08 and 0.04 (from 1.50‰, 0.83‰ and 0.56‰ in 2014 to 1.58‰, 0.91‰ and 0.60‰ in 2015). They coincided with the aged mother being attributable for mortalities in children within 1 year.

2. Page 7 line 15-19. It is a bit confusing in the text. U5 death included Infant death or not? Similarly, infant death include neonate death or not? According to the text, it looks that they are not included to the others (mutually exclusive). However, next sentence "death in neonates accounted for more than 50% of all death among for under-5 children" made audience to think they not mutually exclusive to each other (U5 death include infant death; and infant death include neonate death). Which one is correct? Please be consistent or make it not confusing. New raw for total death next to 2015 will make audience less confusing as well.

Reply: Thanks for your comments.

This comment is similar to the previous comment. We clarified the definition of "neonatal mortality rate (NMR)", "infant mortality rate (IMR)" and "under-5 mortality rate (U5MR)" in methods section (Page 6, Line 2-6). That is to say, U5MR included IMR, and IMR included NMR. What's more, the section in Page 6, Line 20-21 has been corrected into neonatal deaths, infant deaths and under-5 children deaths. We hope that it is not confusing any more this time.

3. Page 8 line 38. "mortality rates for leading cause of death...overall decreasing..." sounds very illogical. Some mortality rates, which were leading cause of death in 1992, can be decreasing since then. However, general decreasing of leading cause of deaths sounds very inappropriate expression. Other cause became major leading causes? Or Same rank of cause of death but all decreasing in terms of mortality per se? Even the latter is right, this section is not the decomposition of all cause mortality, but cause of death distribution mostly based on rank order description. So in this sense, the expression is not suitable to the section heading.

Reply: Thanks for your comments.

We corrected the section heading (Page 7, Line 12), because this section mainly describes the distribution of leading causes of death. The change in rank order and reduction in mortality rates for leading causes of death are described in Table 2, Page 7, Line 13-18 and Page 8, Line 1-5. We hope that it is not confusing any more this time.

4. Page 10 line 3. The percentages look confusing? What do they mean?

Reply: Thanks for your comments.

The three percentages are the largest cause-specific annual rate changes in congenital heart disease (77.08%, from 31.71 in 1997 to 56.15 in 1998 per 100,000 livebirths), other congenital abnormalities (94.90%, from 3.76 in 1999 to 7.34 in 2000 per 100,000 livebirths) and accidental asphyxia (389.90%, from 2.06 in 2001 to 10.11 in 2002 per 100,000 livebirths) in 28 days-1 year children (Page 10, Line 1-5). These results indicated that the cause-specific mortalities in congenital heart disease, other congenital abnormalities and accidental asphyxia had obvious fluctuation, which is different from the trends in other age groups (Figure 2).

5. Upward trend forecasting may be model misspecification due to adapting the function of "y=aX square+bX+c". Inverse root X function or logarithmic function would predict differently.

Reply: Thanks for your comments.

In the present study, we used ARIMA model to forecast the U5MRs using surveillance data. Since U5MR has many aspects of influencing factors such as population, policy, economy and environment and has continuity in time traditional statistical analysis such as regression analysis methods are unsuitable. As it clarifying in the section of "Statistical analysis" (Page 6, Line 11-13), ARIMA model can use time to replace kinds of influencing factors and can be established on the base of past values of series and previous error for forecasting. It has been reported that ARIMA model can be applied to forecast incidence of hand-foot-mouth disease[1], influenza H5N1[2] and injury[3].

The forecasting model ARIMA(1, 1, 1) is: $x_t = -0.445 + 1.509x_{t-1} - 0.509x_{t-2} + \alpha_t - 0.999\alpha_{t-1}$. It is established on the base of past values of series (x_{t-1} , x_{t-2}) and previous error (α_t , α_{t-1}), not the formula of $y = ax^2 + bx + c$ which you mentioned before. Therefore, the forecasting mortalities are predicted according to the previous trend and don't adapt the function any more.

In the model we established, $R^2=0.982$, $AIC=2.15$, $BIC=2.30$ and $MAPE=4.76\%$. Therefore, it is appropriate to forecast U5MRs by ARIMA model.

[1] L. Yu, L. Zhou, L. Tan, H. Jiang, Y. Wang, et al., Application of a new hybrid model with seasonal auto-regressive integrated moving average (ARIMA) and nonlinear auto-regressive neural network (NARNN) in forecasting incidence cases of HFMD in Shenzhen, China. PLoS One, 2014. 9(6): e98241.

[2] M. J. Kane, N. Price, M. Scotch, P. Rabinowitz, Comparison of ARIMA and Random Forest time series models for prediction of avian influenza H5N1 outbreaks. Scientific World Journal, 2014. 15:276.

[3] Y. Lin, M. Chen, G. Chen, X. Wu, T. Lin, Application of an autoregressive integrated moving average model for predicting injury mortality in Xiamen, China. BMJ Open, 2015. 5(12): e008491.

6. Page 14, Line 34-42. It sounds too redundant.

We simplified it (Page 14, Line 22-30 and Page 15, Line 1).

7. Fig 1. Same color should be consistent across A-D so that no confusion rises to the audience.

Purple represented pneumonia in A-D consistently. However, yellow color was for birth asphyxia in A-B but in C, it meant septicemia; in D, it meant leukemia. By this way, chart looks not succinctly show the mortality composition change but making confusion. Please use different color in different cause of death.

We used different colors to represent different causes of death in Fig 2. A-D.

Reviewer: 2

1. Latest advances in predicting mortality increasingly include Bayesian inference and they account for the distribution of the age at death in order to capture detailed patterns of mortality. In addition, some recent studies used rates of improvement or rate changes rather than death rates to forecast mortality changes better. Whether you use classical or Bayesian approach, to address the objectives in your paper more efficiently:

1) It would be better to use forecasting models which accounts for different age groups (infants, neonates, ...).

2) It would be better to forecast mortality for each major cause of death.

3) It would be better to use forecasting models which accounts for other related impact factors.

Reply: Thanks for your comments. It is very constructive.

As the review mentioned above, Bayesian inference is really an excellent method and can be used in many fields. The present study used the surveillance data between 1992 and 2015 from Beijing Under-5 mortality rate Surveillance Network and aimed to analyze the dynamic changes in U5MR.

The index of U5MR is an important indicator which reflects children's health and the development of economy and culture in a country or area. U5MR is different from cause-specific mortality rate, which is affected by many aspects of influencing factors such as population, policy, economy and environment. In addition, we can only acquire death data (case group) from the surveillance network

and lack of healthy children data (control group). Therefore, using traditional regression analysis methods which consider related impact factors to predict is limited.

ARIMA model can use time to replace kinds of influencing factors and can be established on the base of past values of series and previous error for forecasting. It has been applied to forecast incidence of hand-foot-mouth disease[1], influenza H5N1[2] and injury[3]. What's more, U5MR has continuity in time. In the model we established, $R^2=0.982$, $AIC=2.15$, $BIC=2.30$ and $MAPE=4.76\%$. Therefore, it is appropriate to forecast U5MRs by ARIMA model.

Bayesian inference can use apriori information and sample information to establish steadier forecasting model to capture detailed patterns of mortality. Bayesian inference and ARIMA model both have advantages and disadvantages in prediction. We intend to process another further study to compare the effect of Bayesian inference and ARIMA model in predicting U5MRs.

Furthermore, in the present study, we only intended to observe the long-term and prospective changing trend in mortality rates in entire under-5 children through the forecasting model, so we didn't consider to accounts for different age groups and specific diseases. At the same time, considering the length of the article, we are going to consider the suggestions in the future.

[1] L. Yu, L. Zhou, L. Tan, H. Jiang, Y. Wang, et al., Application of a new hybrid model with seasonal auto-regressive integrated moving average (ARIMA) and nonlinear auto-regressive neural network (NARNN) in forecasting incidence cases of HFMD in Shenzhen, China. PLoS One, 2014. 9(6):e98241.

[2] M. J. Kane, N. Price, M. Scotch, P. Rabinowitz, Comparison of ARIMA and Random Forest time series models for prediction of avian influenza H5N1 outbreaks. Scientific World Journal, 2014. 15:276.

[3] Y. Lin, M. Chen, G. Chen, X. Wu, T. Lin, Application of an autoregressive integrated moving average model for predicting injury mortality in Xiamen, China. BMJ Open, 2015. 5(12):e008491.

2. Introduction: Line 13-14 it would be better to rephrase the statement for better meaning.

We did it.

3. Methods: Line 44-46 it would be better to rewrite the statement for better meaning.

We did it.

4. Methods: Line 53-56 check the sentence for coherence or may be missing statement.

We did it.

5. Results: On page 7 line 17 (death in neonates accounted for more than 50% of all deaths...) not coincides with the figure in table 1.

We added Figure 1 to explain.

6. Results: Page 8 line 21-26 it would be better to rephrase for better meaning.

We did it.

7. Results: Page 8 line 38-42 (... Decreased by ...) it is not clear for which time period or relative to what the figure stands for.

We clarified it (Table 2, Page 7, Line 13-18 and Page 8, Line 1-7).

8. Results: Page 8 line 44-49 check for coherence or sentence fragment.

We did it.

9. Results: On page 9 table 2, it would be more informative to put mortality rate for all years. There will be a possibility of higher mortality rate even for those having rank order more than five.

We added it (Table 2).

10. Results: Page 10 line 9-11 it would be better to rephrase the sentence for better meaning.

We did it.

11. Results: Page 10 line 43-48.

We did it.

12. Discussion: Page 12 line 49 better to rewrite the statement.

We did it.

13. Discussion: Page 13 line 13-14 sentence fragment, formal writing.

We did it.

14. Discussion: Page 13 line 39-40 check for sentence fragment.

We did it.

15. Discussion: Page 13 line 47-52 it would be better to rephrase the sentence for better meaning.

We did it.

16. Discussion: Page 14 line 28 rewrite the statement.

We did it.

17. Discussion: Page 14 39-43 check for sentence coherence.

We did it.

18. Discussion: Page 15 line 2-28 As I go through your paper unable to find a model incorporating the factor "Selective Two-child Policy". I wonder if you could justify how you come to this decision.

Reply: Thanks for your comments.

It seems we haven't explained clearly. Our results could not detect the effect of "Selective Two Child Policy" directly and just suggest that the policy regulation may affect slightly upward trend in mortality to rise healthcare workers' attention about this potential problem.

Because we only collected 2-year data after the "Selective Two Child Policy", the representativeness of the changing trend and solid statistical analysis was limited. We will continue collecting the surveillance data to do further analysis to draw more rigorous conclusion about this problem (Page 3, Line 17-20 and Page 15, Line 15-19).

19. ABSTRACT (Methods)

Line 14 (...analyzed using statistical description...) It would be better to rephrase the statement for better meaning.

We did it.

20. ABSTRACT (Results)

Line 21 sentence defragment.

We did it.

Line 21-31 inconsistencies such as: to vs - and sentence fragment.

We did it.

21. ABSTRACT (Conclusions)

Line 36 (...U5MRs may show a slightly upward trend from 2016 to 2020 as influenced by the "Selective Two-child Policy"...). As I go through your paper unable to find a model incorporating the factor "Selective Two-child Policy". I wonder if you could justify how you come to this decision.

It is similar to Question 18. We could not come to this conclusion and deleted it.

Reviewer: 3

1. The abstract needs to be rephrased. For example, page 2 line 21: please specify the corresponding periods for the three decreasing rates; Several syntax errors.

We did it.

2. Page 4 line 8, line 11, line 19, please add references.

We added Ref 1, Ref 3 and Ref 6.

3. Ref 7, needs to be updated by:

He, Chunhua, et al. "National and subnational all-cause and cause-specific child mortality in China, 1996–2015: a systematic analysis with implications for the Sustainable Development Goals." *The Lancet Global Health* (2016).

Song, Peige, et al. "Causes of death in children younger than five years in China in 2015: an updated analysis." *Journal of global health* 6.2 (2016).

We added them as Ref 3 and Ref 6. They are very constructive.

4. Please change the glossary and specific diseases and to be in accordance with the abovementioned two national studies.

We did it.

5. The writing should be improved with assistance of English native speakers.

We have invited an English native speaker to modify the manuscript.

6. Data between 1992 and 2002 was collected using "double entry and consistency check", is it the same for the "electronic cards" between 2003 and 2015? Please clarify.

Reply: Thanks for your comments. It is very constructive.

We clarified in Page 5, Line 23-26. Child Death Reporting Card was in paperboard from 1992 to 2002, so we applied double entries and consistency checks in the Epidata 3.0 software to make sure the accuracy of the data. However, data collected between 2003 and 2015 was directly exported because electronic cards have been used since the information system was established in 2003 which could ensure the accuracy of the data.

7. Reference the "MCMS Child death report card" and all disease definitions in the Online Supplementary Document of "Causes of death in children younger than five years in China in 2015: an updated analysis" (<https://www.ncbi.nlm.nih.gov/pmc/articles/PMC5140075/#S1>)

We corrected according to it. It is very constructive.

8. Table 1, clarify the unit of "Mortality rate";

We did it (Page 7, Line 1-3).

9. Table 2, add the proportions of death causes, not only the rank orders; And rephrased the corresponding results accordingly.

We added it (Table 2).

10. Page 12, line 39-32 "Deaths due to birth asphyxia declined steadily through Beijing intensifying intrapartum monitoring and promotion of neonatal recovery technology", please discuss if caesarean section plays a role in the decline of deaths due to birth asphyxia.

We added it (Page 13, Line 16-24). It is very constructive.

11. Page 12, line 51-52: "Several studies have verified that congenital heart disease has several influence factors, such as increased maternal age, prenatal infections, gestational hypertension, gestational diabetes, etc." This kind of discussion is hand-waving, further in-depth discussion is needed.

We did it (Page 13, Line 25-30 and Page 14, Line 1-5). It is very constructive.

12. Page12, line 58; page 13 line 4-11: please discuss if caesarean section plays a role.

We did it (Page 14, Line 6-13). It is very constructive.

13. Throughout the discussion part, DO NOT only cite other researchers' statements as explanations of changes of death causes, explore possible reasons in BEIJING-CONTEXT, this is of more interest for readers, and more useful for policy-makers.

Thanks for your comments. It is very constructive. We explored possible reasons in Beijing-context in the section of "Discussion" and marked them in blue color.

VERSION 2 – REVIEW

REVIEWER	Melkamu Dedefo Gishu Haramaya University, Ethiopia University of KwaZulu-Natal, South Africa
REVIEW RETURNED	28-Apr-2017

GENERAL COMMENTS	Thank you for going through the comments and related amendments.
--